# Help-seeking for genitourinary symptoms: a mixed methods study from Britain's Third National Survey of Sexual Attitudes and Lifestyles (Natsal-3)

Fiona Mapp ,[1,2] Kaye Wellings,[2] Catherine H Mercer,[1] Kirstin Mitchell,[3] Clare Tanton ,[4] Soazig Clifton,[1] Jessica Datta,[2] Nigel Field,[1] Melissa J Palmer,[5] Ford Hickson[2]

[1]Institute for Global Health, University College London, London, UK
[2]Department of Public Health, Environments and Society, London School of Hygiene & Tropical Medicine, London, UK
[3]MRC/CSO Social & Public Health Sciences Unit, University of Glasgow, Glasgow, UK
[4]Global Health and Development, London School of Hygiene & Tropical Medicine, London, UK
[5]Department of Population Health, London School of Hygiene & Tropical Medicine, London, UK

**Correspondence to**
Dr Fiona Mapp;
f.mapp@ucl.ac.uk

## ABSTRACT

**Objectives** Quantify non-attendance at sexual health clinics and explore help-seeking strategies for genitourinary symptoms.

**Design** Sequential mixed methods using survey data and semistructured interviews.

**Setting** General population in Britain.

**Participants** 1403 participants (1182 women) from Britain's Third National Survey of Sexual Attitudes and Lifestyles (Natsal-3; undertaken 2010–2012), aged 16–44 years who experienced specific genitourinary symptoms (past 4 weeks), of whom 27 (16 women) who reported they had never attended a sexual health clinic also participated in semistructured interviews, conducted May 2014–March 2015.

**Primary and secondary outcome measures** From survey data, non-attendance at sexual health clinic (past year) and preferred service for STI care; semistructured interview domains were STI social representations, symptom experiences, help-seeking responses and STI stigma.

**Results** Most women (85.9% (95% CI 83.7 to 87.9)) and men (87.6% (95% CI 82.3 to 91.5)) who reported genitourinary symptoms in Natsal-3 had not attended a sexual health clinic in the past year. Around half of these participants cited general practice (GP) as their preferred hypothetical service for STI care (women: 58.5% (95% CI 55.2% to 61.6%); men: 54.3% (95% CI 47.1% to 61.3%)). Semistructured interviews elucidated four main responses to symptoms: not seeking healthcare, seeking information to self-diagnose and self-treat, seeking care at non-specialist services and seeking care at sexual health clinics. Collectively, responses suggested individuals sought to gain control over their symptoms, and they prioritised emotional reassurance over accessing medical expertise. Integrating survey and interview data strengthened the evidence that participants preferred their general practitioner for STI care and extended understanding of help-seeking strategies.

**Conclusions** Help-seeking is important to access appropriate healthcare for genitourinary symptoms. Most participants did not attend a sexual health clinic but sought help from other sources. This study supports current service provision options in Britain, facilitating individual autonomy about where to seek help.

### Strengths and limitations of this study

► We used a sequential mixed methods design to explain and expand survey data about genitourinary symptom experiences and help-seeking behaviour using semistructured interviews.
► We sampled participants for the semistructured interviews from survey participants, reflecting diversity of symptom experiences, personal characteristics and geographical location in order to examine help-seeking independently of medical settings.
► To maximise the value of conducting follow-up interviews, we first undertook survey analysis, to inform questions to explore qualitatively which resulted in a delay of 22–44 months between data collection phases.

## INTRODUCTION

Help-seeking is a complex process defined by Fortenberry[1] as the 'interval between recognition of a health problem and its clinical resolution and… the accompanying cognitive and behavioural responses'. Help-seeking for symptoms relies on individuals interpreting physical sensations and navigating the health system available to them.[2] In Britain sexual health clinics (SHCs), also called genitourinary medicine (GUM) clinics are specialised services within the National Health Service for managing genitourinary health including sexually transmitted infection (STI) testing, diagnosis and treatment and providing sexual health advice. SHCs are accessible without referral from another healthcare professional, open to everyone regardless of nationality or residency status and tend to be located in urban areas (although many operate outreach programmes or basic sexual healthcare provision in more rural areas).[3] SHCs offer greater expertise and better testing options than primary care and do not charge

patients for the care they receive. Treatment is also free unlike medication prescribed from other NHS services (eg, from GPs).[4] Recent funding cuts have reduced the availability of booked and walk-in appointments, have led to some clinic closures and resulted in more asymptomatic patients being managed through online self-sampling pathways.[3] SHCs are sometimes preferred to general practice because they allow patients anonymity as medical histories are not linked to GP or other health records; however, they can also be stigmatised environments.[5 6] While GPs can manage genitourinary symptoms, many lack specialist training, worry about discussing sensitive subject matter and experience time constraints with shorter appointments.[4]

Help-seeking in response to genitourinary symptoms can reduce unmet need and untreated infection. SHC attendance has increased over the last three decades[7] with symptoms being the most commonly reported reason for attendance in England.[8 9] However, while 21% of women and 6% of men are estimated to have experienced genitourinary symptoms in the past month[10] (equating to almost 3.3 million adults in Britain), national surveillance data recorded 2 million attendances (excluding follow-up attendances) at SHCs in England in 2016.[11] This suggests that a proportion of people with symptoms do not attend SHCs.

Genitourinary symptoms, such as painful urination and abnormal vaginal or penile discharge, can indicate underlying infections or disease such as those that are sexually transmitted.[12] If left undiagnosed and untreated, underlying disease can cause serious harm to individuals and, in the case of STIs, their sexual partners.[12] This lack of response to symptoms contributes to the burden of poor sexual and reproductive health in the population and reduces individual quality of life and well-being. Effective and timely treatment is important in mitigating deleterious effects of STIs and other causes of genitourinary symptoms for individual and population health. There is, however, currently little evidence about help-seeking among people with genitourinary symptoms,[13] especially choices that do not involve visiting health services. Non-attendance is irrational from a medical perspective but may be rational for individuals depending on their subjective values and beliefs about health and healthcare[14] (eg, to avoid stigmatisation).

In this paper, we use genitourinary symptoms as an indicator of potential need for care and draw on survey and semistructured interview data from the Third National Survey of Sexual Attitudes and Lifestyles (Natsal-3) to understand reasons for non-attendance at SHCs and explore help-seeking strategies in response to symptoms.

## METHODS
### Study design
Full details of methods are described in the published study protocol.[15] Briefly, we combined survey data and data from follow-up semistructured interviews to connect, explain and extend findings about help-seeking for genitourinary symptoms. Following preliminary analysis of data from the Natsal-3 survey, we used survey participants' responses relating to experience of symptoms and non-attendance at SHCs to draw a subsample invited to participate in follow-up semistructured interviews. Data from the entire Natsal-3 survey were used to contextualise interview data and we integrated findings from the two datasets to provide combined insights into help-seeking strategies for symptoms.

### Natsal-3 survey
Natsal-3 is a probability sample survey (n=15 162) of sexual behaviour among women and men resident in Britain aged 16–74 years[16] with 58% response rate. Interviews used computer-assisted personal interview and computer-assisted self-interview (CASI) for sensitive topics. In the CASI, sexually experienced participants (defined as having reported at least one lifetime sexual partner) aged 16–44 years were asked about genitourinary symptoms (see box 1). The list of symptoms are routinely asked about in sexual health consultations.

We calculated the prevalence of non-attendance at SHCs in the past year among those who reported symptoms as an indicator of potential unmet need for healthcare. We then examined hypothetical service preferences (see box 1). We used logistic regression to calculate ORs for stating SHC, adjusting for previous SHC attendance. Analyses were carried out using survey commands in Stata V.14.1 to account for stratification, clustering and weighting of survey data and were stratified by gender to reflect differences in reported care-seeking behaviour,[17] symptom prevalence and emergent findings from semistructured interviews.

### Semistructured interviews
We wanted to examine the reasons for SHC non-attendance, so we explored help-seeking responses to experiencing genitourinary symptoms. Participants who had agreed to be recontacted, had reported symptom(s) and had never attended an SHC were recruited for a face-to-face semistructured interview (conducted by FM) at their home or other convenient location. We used purposive sampling from eligible survey participants to reflect diversity of personal characteristics (age and sex), genitourinary symptom experiences and geographical location (rural/urban/metropolitan settings and different areas of Britain) among participants. Interviews took place between 22 and 44 months (median=30 months) after the fieldwork for Natsal-3 was conducted. The delay in conducting interviews enabled initial survey analyses to be carried out to inform the topic guide and focus on explaining non-attendance at SHC. Interviews lasted between 35 and 108 min, and participants received a £20 shopping voucher on completion of the interview. FM wrote field notes after each interview and discussed these with FH and KW to encourage reflexive practice.

## Box 1  Survey question wording and response options

**'In the last month, that is since (date one month ago), have you had any of the following symptoms?'**
Response options:
Women:
1. Pain, burning or stinging when passing urine.
2. Passing urine more often than usual.*
3. Genital wart/lump.
4. Genital ulcer/sore.
5. Abnormal vaginal discharge.
6. Unpleasant odour associated with vaginal discharge.
7. Vaginal pain during sex.
8. Abnormal bleeding between periods.
9. Bleeding after sex (not during a period).
10. Lower abdominal or pelvic pain (not related to periods).
11. None of these.
Men:
1. Pain, burning or stinging when passing urine.
2. Passing urine more often than usual.*
3. Genital wart/lump.
4. Genital ulcer/sore.
5. Discharge from the end of the penis.
6. Painful testicles.
7. None of these.
*Excluded in this study following advice and discussion with clinical Natsal-3 members as more indicative of urinary tract infections, not STIs.

**'*If* you thought that you might have an infection that is transmitted by sex, where would you *first* go to seek diagnosis and/or treatment?'**
Response options:
1. General practice surgery,
2. Sexual health clinic (GUM clinic).
3. National Health Service (NHS) Family planning clinic/contraceptive clinic/reproductive health clinic.
4. NHS antenatal clinic/midwife.
5. Private non-NHS clinic or doctor.
6. Pharmacy/chemist.
7. Internet site offering treatment.
8. Youth advisory clinic (eg, Brook clinic).
9. Hospital accident and emergency (A&E) department.
10. Somewhere else.

Interviews were digitally recorded and transcribed verbatim. We used principles of Interpretative Phenomenological Analysis[18 19] to explore lived experiences and meanings of help-seeking strategies in response to symptoms. Data were coded case by case, and emergent themes were grouped to identify connections within and between transcripts. Through discussion with other authors, we refined themes and gained different perspectives on the data. We organised the data into different help-seeking pathways as the explanations for non-attendance at SHCs and explored themes within and across each pathway to understand how individuals had made sense of their care needs. We further classified data according to whether participants described symptoms they had reported in the survey, additional symptoms or different symptoms.

We used NVivo V.11 to organise data, and one-third of transcripts were double coded by KW and FH.

### Data integration
We used a convergence coding matrix[20] to integrate survey and semistructured interview data by research theme and move beyond the method through which data were generated to become more conceptual ideas and gain a more complete picture of help-seeking. After analysing survey and interview data separately, we presented key data relating to each theme side by side in the matrix to look for areas of agreement, contradiction and silence.[21] FM and FH conducted the integration through discussion of each theme in turn and added findings into the last column of the matrix.

### Patient and public involvement
Patients or members of the public were not involved in the development, design or conduct of this study.

## RESULTS
We present survey data first to quantify non-attendance at SHCs and other service preferences, followed by semistructured interview data to broaden analyses to understand the reasons for non-attendance behaviour and other help-seeking strategies.

### Survey data
#### Participants
Detailed descriptions of the Natsal-3 sample have already been reported.[22] Of all sexually experienced participants aged 16–44 years (unweighted n=8878; weighted n=7353), 21.6% (95% CI 20.4% to 22.9%) of women and 5.6% (95% CI 4.9% to 6.6%) of men reported recent (past 4 weeks) genitourinary symptoms. Data were missing among 1.4% for reported symptoms and 3.4% for reported SHC attendance.

#### Non-attendance at SHCs
The prevalence of non-attendance at an SHC in the past year for all women and men reporting recent symptoms was high (women: 85.9% (95% CI 83.7% to 87.9%); men: 87.6% (95% CI 82.3% to 91.5%)). There were no significant gender differences in attendance behaviour (see table 1). We found higher levels of non-attendance with increasing age for both women and men. We examined never attending SHCs among those reporting symptoms and found that 55.8% (95% CI 52.5% to 59.1%) of women and 53.8% (95% CI 46.2% to 61.2%) of men had never attended.

#### Service preference
General practice was the preferred provider for hypothetical STI care for both women (58.5%, 95% CI 55.2% to 61.6%) and men (54.3%, 95% CI 47.1% to 61.3%) who reported symptoms (table 2). Participants with symptoms who had previously attended an SHC were more likely to choose an SHC as their preferred hypothetical service

**Table 1** Prevalence of reported non-attendance at an SHC in the past year among sexually experienced participants aged 16-44 years who reported recent symptoms by age group and sex

| Age group (years) | Women | | Men | |
| --- | --- | --- | --- | --- |
| | Non-attendance in past year % (95% CI) | Denominator*: unweighted, weighted | Non-attendance in past year % (95% CI) | Denominator*: unweighted, weighted |
| 16–24 | 73.6 (69.0 to 77.8) | 474, 268 | 78.2 (67.7 to 86.0) | 98, 70 |
| 25–34 | 89.4 (86.2 to 92.0) | 518, 305 | 88.9 (79.5 to 94.3) | 84, 77 |
| 35–44 | 95.9 (91.8 to 97.9) | 190, 222 | 97.0 (88.5 to 99.3) | 39,† 61 |
| All ages | 85.9 (83.7 to 87.9) | 1182, 795 | 87.6 (82.3 to 91.5) | 221, 208 |
| P value‡ | <0.0001 | | 0.0014 | |

*Denominator is all sexually experienced women and men aged 16–44 years who reported symptoms; excludes participants with missing data for symptom variables.
†Small number of participants so estimates may be unreliable.
‡χ2 p value for association with age group.
SHC, sexual health clinic.

than those who had not previously attended an SHC (women: 57.7% (95% CI 53.0% to 62.3%) vs 14.8% (95% CI 11.7% to 18.5%), age-adjusted OR 7.3 (95% CI 5.3 to 10.0); men: 63.8% (95% CI 53.0% to 73.4%) vs 19.7% (95% CI 13.1% to 28.5%), age-adjusted OR 7.2 (95% CI 3.6 to 14.2), data not shown).

### Semistructured interview data
#### Participants
Semistructured interviews were completed with 27 Natsal-3 participants: 16 women and 11 men, aged 19–47 years. The majority were white British/other white, four were Asian/Asian British or black/black British; five did not have English as their first language but were sufficiently fluent to participate in an English-language interview. Participants' lifetime experiences of genitourinary symptoms and help-seeking are described in table 3. Help-seeking varied between participants and by symptom(s).

### Explanations for reported non-attendance at SHCs by recently symptomatic survey participants
Survey data suggested that it was common for symptomatic participants to not attend an SHC. It also showed the GP was the preferred hypothetical care provider. Our semistructured interview findings generated several explanations for non-attendance at clinics and preference for non-specialist care: not seeking healthcare, seeking information to self-diagnose and/or self-treat, seeking care at a non-specialist sexual health service and those who reported seeking care at an SHC. These are discussed separately and then interpreted collectively as seeking control over symptom experiences. Data are integrated in table 4.

#### Not seeking healthcare
Individuals were highly selective about which symptoms they responded to, resulting in many symptoms not being

**Table 2** Hypothetical service choice of sexually experienced participants aged 16–44 years who reported symptoms stratified by sex and age group

| Age group (years) | Women % (95% CI) | | | Men % (95% CI) | | |
| --- | --- | --- | --- | --- | --- | --- |
| | 16–24 | 25–34 | 35–44 | 16–24 | 25–34 | 35–44 |
| GP | 44.5 (39.5–49.6) | 58.4 (53.5–63.2) | 75.4 (68.1–81.5) | 53.7 (42.8–64.3) | 38.29 (27.7–50.2) | 75.96 (56.6–88.4) |
| SHC | 43.6 (38.6–48.7) | 35.7 (31.0–40.6) | 19.3 (14.0–26.2) | 38.91 (28.9–50.0) | 52.9 (41.0–64.4) | 24.04 (11.6–43.4) |
| Other† | 12.0 (9.1–15.6) | 5.9 (4.0–8.6) | 5.3 (2.7–10.0) | 7.4 (3.3–15.7) | 8.8 (3.4–21.1) | 0 |
| Denominator†: weighted, unweighted | 268, 474 | 305, 518 | 222, 190 | 70, 98 | 77, 84 | 59, 38‡ |

*Other healthcare services: NHS family planning clinic/contraceptive clinic/reproductive health clinic; NHS antenatal clinic/midwife; private non-NHS clinic or doctor; pharmacy/chemist; internet site offering treatment; youth advisory clinic (eg, Brook clinic); hospital accident and emergency department; and somewhere else.
†Denominator is all sexually experienced women and men aged 16–44 years who reported symptoms.
‡Small numbers, therefore estimates may be unreliable.
GP, general practice; SHC, sexual health clinic.

**Table 3** Overview of qualitative participants' reported genitourinary symptoms, hypothetical service preference and care-seeking behaviour

| Interview no. | Sex | Age* | Symptoms reported in the Third National Survey of Sexual Attitudes and Lifestyles (Natsal-3) (past month) | Symptoms reported in semistructured interview (ever) | Hypothetical service preference | Care-seeking for symptoms reported in semistructured interview (ever) |
|---|---|---|---|---|---|---|
| Data source | CAPI | CAPI, SSI | CASI | SSI | CASI | SSI |
| i2 | Female | 35–39 | Abdominal/pelvic pain. | Pain urinating; vaginal pain during sex; bleeding after sex; and abdominal/pelvic pain. | GP | GP for abdominal pain, referred on to NHS gynaecologist. |
| i3 | Female | 20–24 | Abdominal/pelvic pain. | Abnormal vaginal discharge; vaginal pain during sex; and abdominal/ pelvic pain. | SHC | GP and private gynaecologist for different symptoms. |
| i4 | Female | 25–29 | Abnormal bleeding between periods; and abdominal/pelvic pain. | Pain urinating; abnormal vaginal discharge; vaginal pain during sex; abnormal bleeding between periods; bleeding after sex; and abdominal/pelvic pain. | SHC | None. |
| i6 | Female | 35–39 | Abnormal bleeding between periods. | Pain urinating and abnormal vaginal discharge. | SHC | Cannot remember. |
| i7 | Female | 40–44 | Genital ulcer/sore. | Pain urinating; genital ulcer/ sore; abnormal vaginal discharge; and abnormal bleeding between periods. | GP | None. |
| i8 | Female | 16–19 | Abnormal bleeding between periods. | Pain urinating; vaginal pain during sex; abnormal bleeding between periods; bleeding after sex; and abdominal/pelvic pain. | FPC | GP for abnormal bleeding between periods and abdominal pain. |
| i9 | Female | 20–24 | Pain urinating; vaginal pain during sex; and abnormal bleeding between periods. | Pain urinating; abnormal vaginal discharge; unpleasant odour associated with vaginal discharge; vaginal pain during sex; and abnormal bleeding between periods. | FPC | SHC for abnormal vaginal discharge and abnormal bleeding between periods. |
| i10 | Male | 20–24 | Painful testicles. | Painful testicles. | GP | None. |
| i11 | Male | 16–19 | Painful testicles. | None. | GP | None. |
| i12 | Female | 25–29 | Unpleasant odour associated with vaginal discharge. | Pain urinating; abnormal vaginal discharge; unpleasant odour associated with vaginal discharge; and abdominal/pelvic pain. | GP | GP for abnormal discharge and odour, referred to hospital for further investigations; and midwife for abdominal pain during pregnancy. |
| i13 | Male | 20–24 | Genital wart/lump. | Genital wart/lump. | SHC | SHC (different town) after third episode of warts. |
| i14 | Male | 45–49 | Pain urinating. | Pain urinating; genital lump (not a wart) and painful testicles. | GP | GP for lump in testicles. |
| i15 | Male | 30–34 | Pain urinating. | Pain urinating and painful testicles. | GP | Pharmacist for pain urinating. |
| i16 | Female | 25–29 | Abdominal/pelvic pain. | Pain urinating; abnormal vaginal discharge; abnormal bleeding between periods; and abdominal/ pelvic pain. | FPC | GP for all symptoms except discharge and pharmacist for thrush (self-diagnosed). |
| i17 | Male | 30–34 | Penile discharge. | Pain urinating; penile discharge; and painful testicles. | SHC | None. |
| i18 | Female | 30–34 | Pain urinating. | Pain urinating and unpleasant odour associated with vaginal discharge. | GP | GP for all symptoms. |

Continued

**Table 3** Continued

| Interview no. | Sex | Age* | Symptoms reported in the Third National Survey of Sexual Attitudes and Lifestyles (Natsal-3) (past month) | Symptoms reported in semistructured interview (ever) | Hypothetical service preference | Care-seeking for symptoms reported in semistructured interview (ever) |
|---|---|---|---|---|---|---|
| Data source CAPI | | CAPI, SSI | CASI | SSI | CASI | SSI |
| i19 | Female | 30–34 | Bleeding after sex and abdominal/pelvic pain. | Pain urinating; abnormal vaginal discharge; vaginal pain during sex; abnormal bleeding between periods; and abdominal/pelvic pain. | SHC | Mentioned abnormal bleeding at contraception clinic visit but no care-seeking specifically for symptoms. |
| i20 | Male | 30–34 | Pain urinating and painful testicles. | Pain urinating; penile discharge; painful testicles. | GP | GP for all symptoms. |
| i21 | Male | 20–24 | Painful testicles. | None. | FPC | None. |
| i22 | Male | 30–34 | Painful testicles. | Pain urinating and painful testicles. | GP | GP for both symptoms. |
| i23 | Male | 20–24 | Painful testicles. | Pain urinating and painful testicles. | GP | GP for both symptoms. |
| i24 | Male | 16–19 | Painful testicles. | Pain urinating and painful testicles. | GP | SHC for pain urinating and GP for painful testicles |
| i25 | Female | 45–49 | Unpleasant odour associated with vaginal discharge. | Pain urinating and abnormal vaginal discharge. | GP | GP for both symptoms. |
| i26 | Female | 16–19 | Genital ulcer/sore. | Pain urinating; abnormal vaginal discharge; unpleasant odour associated with vaginal discharge; and vaginal pain during sex. | SHC | Went to hospital for pain urinating. |
| i27 | Female | 25–29 | Genital ulcer/sore and genital wart/lump. | Pain urinating; abnormal vaginal discharge; vaginal pain during sex; and abdominal/pelvic pain. | GP | GP for pain urinating and midwife for abdominal pain (during pregnancy). |
| i28 | Female | 30–34 | Unpleasant odour associated with vaginal discharge. | Abnormal vaginal discharge. | GP | None. |
| i29 | Female | 40–44 | Unpleasant odour associated with vaginal discharge. | Pain urinating; abnormal vaginal discharge; unpleasant odour associated with vaginal discharge odour; and abdominal/pelvic pain. | Internet | GP and private gynaecologist. |

Shaded columns contain data from Natsal-3 survey.

*Age at time of qualitative interview is calculated using the participant's date of birth and date of follow-up interview;.

CAPI, computer-assisted personal interview; CASI, computer-assisted self-interview; FPC, family planning clinic/contraceptive clinic/reproductive health clinic; GP, general practitioner (primary care); GUM, genitourinary medicine; Natsal-3, third National Survey of Sexual Attitudes and Lifestyles; SHC, sexual health/GUM clinic; SSI, semistructured interview.

presented to a healthcare professional. A quarter of participants reported not seeking care from any health service in response to experiencing symptoms. Instead participants responded by concealing symptoms, normalising them as physiological fluctuations or dismissing any care needs. STI stigma was a factor for many participants who chose not to seek healthcare, and real or perceived structural barriers around accessing services were also cited as reasons for not seeking help or not attending care.

*Concealment of symptoms*

Symptoms were concealed through non-disclosure or partial disclosure (to chosen individuals and/or for specific symptoms). For example, women explained that vaginal discharge was rarely discussed with others as it was seen as too personal and not acceptable to talk about with friends or family members. Concern over what others would think discouraged many from disclosing their experiences. These decisions were presented as rational and considerate about not '*want[ing] to put that burden on anybody*' (i14). Participants also articulated uncertainty about how people would react and so non-disclosure helped to minimise or prevent potential social judgement directed at individuals with symptoms. Multiple examples of non-disclosure and fear of judgement from friends,

**Table 4** Convergence coding matrix: integration of findings from quantitative and qualitative strands according to research themes

| Theme | Quantitative findings | Qualitative findings | Integration |
|---|---|---|---|
| Engagement with SHCs | ► High levels of non-attendance at SHCs for symptomatic women and men in the past year although approximately half had been to an SHC before.<br>► Younger people more likely to have attended than older people.<br>► No significant gender differences in attendance. | ► Some younger participants had attended SHCs for symptoms and STI testing (delays in help-seeking and misreporting in survey).<br>► Most participants did not think their symptoms were caused by STIs so did not seek specialist care at SHCs.<br>► Younger participants were more aware of SHCs. | ► Use of SHCs can vary depending on type of symptoms experienced and perceived cause of symptoms.<br>► SHCs perceived as a service for younger people.<br>► Qualitative findings help explain quantitative data. |
| Service preference | ► GP preferred unless individuals had previously attended an SHC. | ► GPs were a more familiar, less stigmatised type of healthcare service because of their generalist approach.<br>► Some participants preferred the specialism of SHCs once they were familiar with the service. | ► Decision making about care needs and care-seeking is often complex.<br>► Choice of different services valued.<br>► Need to better understand links between hypothetical service preferences and actual care-seeking behaviour for genitourinary symptoms.<br>► Qualitative findings help explain quantitative data. |
| Use of alternative services | No quantitative data. | ► Did not seek any healthcare: concealment, normalisation and dismissal.<br>► Sought information (internet and social network) to self-diagnose/self-treat.<br>► Sought care at another service: mainly GP. | N/A – qualitative data provided exploratory insight into this area. |

GP, general practice/practitioner; SHC, sexual health clinic; STI, sexually transmitted infection.

family and health professionals suggest that stigma is an implicit factor influencing non-healthcare-seeking behaviour. As genitals are generally covered up, it was easy for most participants to conceal their physical symptoms from others day to day. Some symptoms resulted in socially discernible clues, such as *'going to the toilet all the time'* (i6), *'touch[ing] your genitals when you sit down to find a comfortable position'* (i22) or *'not going out'* (i24), which made concealment more difficult. Concealing symptoms from sexual partners often involved abstaining from sex. Some participants mentioned washing more frequently to try and 'get rid' of symptoms, particularly vaginal and penile discharge.

There were individuals who had not told anyone about their symptoms, until they reported them in Natsal-3. The semistructured interview was the first opportunity participants had to describe their experiences.

FM: And did you tell anyone about it [penile discharge]?

Participant: No, I didn't. No, I must admit I didn't even tell my wife, just kept it [penile discharge] private, kept it to myself, just kept looking every day and hoping it would [disappear] … I didn't go to the doctors, I didn't even Google it to be fair, I just hoped it would go away. (i17, man, 30–34 years)

Concealing symptoms from others eliminated social expectations about appropriate care-seeking behaviours, perpetuating non-attendance. Concealment suggests that individuals would prefer to deal with the personal and health consequences of their symptoms than the social consequences of disclosing to others.

### Normalising symptoms and care-seeking behaviours

Normalising symptoms as natural bodily changes, especially by women, eliminated perceived need for any type of care, resulting in non-attendance at services. Social norms about certain symptom experiences, such as painful testicles for men and bleeding problems for women, suggested these issues were *'quite a common thing'* (i26) and not associated with help-seeking.

They're normal things that every woman would go through really, like the bleeding or the pains and stuff…being sore or having a lump. (i27, woman, 25–29 years)

Participants resisted medicalising their experience and did not consider symptoms to be related to STIs. Recurrent or persistent symptoms increased familiarity and normalised the experience, reducing the likelihood of care-seeking if the experience was not perceived to be having detrimental effects. Similarly, long-term conditions that participants may have sought care for previously did not warrant further help-seeking. Instead, participants *'had to get on with it'* (i12) and accepted symptoms as part of their lived reality and sense of self, reducing the impetus to act.

### Dismissal of healthcare needs

Many participants' accounts reflected dismissal of a need to seek care. Some experiences were seen as *'not something you sort of go to your doctors with'* (i4) suggesting

the relationship between experiencing symptoms and seeking care was not a simple causal sequence. In such cases, symptoms were perceived as mild and participants dismissed care-seeking as '*wasting their [doctor's] time*' (i12). Beliefs about the responsible use of healthcare came out particularly strongly in accounts of those who did not seek care for their symptoms, behaviour that affirmed a self-perceived identity as a responsible healthcare user. Participants made care-seeking decisions that were appropriate and rational to them, based on their previous experiences of symptoms and perceived severity, which often resulted in non-attendance at SHCs.

Women in particular did not see the need for healthcare if symptoms related to their sexual activity. There were clear distinctions made between '*medical issues*', which *occurred within* the female body that could be addressed through biomedical intervention, and sexual problems, which were *endured by* the female body and considered to be personal and private matters. Symptoms related to sex, such as pain during and bleeding afterwards, were rarely reported to healthcare professionals.

> I wouldn't go to the doctors because I think that everybody's different in that sense and I don't find it as a medical thing where there might be something medically wrong or I might be ill or there might be a fault. (i27, woman, 25–29 years)

Participants did not seek medical solutions for symptoms related to sex and managed them within their sexual partnerships. The majority of participants did not link their symptoms with STIs. Participants were keen to avoid being diagnosed with an STI as that would '*make me feel a bit dirty, it would make me feel a bit stupid… and I'd panic because I don't know anything about it*' (i9). Dismissal of potential needs and avoiding interactions with healthcare minimised this risk.

### Seeking information to self-diagnose and/or self-treat

For participants who did interpret symptoms as a health problem but did not actually seek medical care, self-diagnosis and self-treatment were common responses. We found several examples of participants attributing their symptoms to other conditions (particularly pain urinating as a UTI and vaginal discharge as thrush). Individuals were reliant on the internet, their social networks and previous experience of the same or similar symptoms to diagnose themselves. Immediacy and convenience of information were frequently prioritised over accuracy.

> Trying to get into the doctors is hell sometimes, being told you've got three weeks to wait for an appointment when you've got all these symptoms busting out… so it's more convenient to just Google it and self-diagnose, even if you've been diagnosed for the wrong thing. (i16, woman, 25–29 years)

Self-diagnosis gave individuals an explanation they could act on to manage their symptoms. Accounts of self-treatment were common and took two forms:

buying over-the-counter medication (general analgesics or specific treatments for thrush or cystitis) and dietary changes such as drinking cranberry juice, reducing alcohol intake and increasing water consumption. Information from Google and advice from friends and family helped guide subsequent decisions about seeking care from healthcare services if self-care options did not resolve the issue (although care-seeking outcomes varied substantially; see table 3). Care-seeking was often based on the experiences and care pathways of their social network and was often influenced by structural factors, particularly those related to service accessibility: location, appointment availability and perceived ease of access. Seeking emotional reassurance from others' lived experiences (online and in real life) was prioritised over biomedical information by many participants.

### Seeking care at a non-specialised sexual health service

Sixteen of 27 participants reported they sought care at a service other than an SHC for their symptoms, and more than half had consulted their GP about their symptoms (table 3). These findings supported service preferences observed in survey data. Presenting symptoms to a GP removed the necessity to navigate unfamiliar parts of the healthcare system, once the need for care had been established; one participant stated that '*if you don't know you've got the symptoms for that particular disease, you don't know to go to a sexual health clinic*' (i11). Some participants relied on their GP to legitimise their need for specialist care, another manifestation of wanting to be a responsible patient, although this often added in an additional care-seeking process and potential delay to receiving treatment.

Women were better linked in to a local network of healthcare services than men through accessing contraception, smear tests, pregnancy care and other gynaecological healthcare. Engagement with familiar healthcare services provided opportunities to discuss genitourinary symptoms and gain access to treatment and reassurance even if they had not specifically sought care for their symptoms. The general nature of non-SHC services offered individuals anonymity regarding their healthcare needs. SHCs differed from other services as participants felt they were labelled as having '*caught something*' as soon as they entered the vicinity of the clinic, making them more vulnerable to social judgement and therefore less likely to seek care at specialised services.

> A lot of people including myself still haven't gone to the clinic because if you're seen outside they go, 'dirty little bitch!'… I had people staring, in the end I went to me doctors. (i16, woman, 25–29 years)

Clinic waiting rooms were perceived to be difficult social environments to negotiate due to stigma associated with STIs, clinics and being seen by others. There were concerns about being judged by other attendees as well as the risk of seeing someone you knew. Clinics were generally unfamiliar environments and represented too many

psychological barriers to overcome to be the preferred choice for care, although after attending once, some of these barriers were removed.

### Seeking care at an SHC

Three participants, all aged under 25 years, attended an SHC in response to their symptoms. They were all very positive about their experiences, valuing the ease of access and specialism. Two other women (both aged 20–25 years) mentioned attending an SHC for STI testing but not in response to having symptoms. These attendance patterns highlight disparities between survey and interview data. Natsal-3 did not capture intention to seek care, and their attendances at SHCs may have occurred after Natsal-3 data collection. There is also concordance with increased likelihood of choosing an SHC having previously attended. There was confusion about the different names and designation of service provision at an SHC and so some misreporting of experience may have occurred in the survey data.

Delays in care-seeking were commonly described, ranging from a few days to several months between the onset of symptoms and attending a healthcare service.

> Yeah, there was a delay… it wasn't straight to the clinic, it was straight to the clinic on the third occasion [of genital warts]… initially there was a two month delay… I was single at the time, the first time it [genital warts] happened, so I wasn't in a rush and I wasn't sexually promiscuous either so I wasn't in a rush to get rid of it. (i13, man, 20–24 years)

In this case, Natsal-3 survey data were collected during or soon after the participant had experienced genital warts but before he had sought care. The timing of the semistructured interview enabled exploration of the participant's story of delayed attendance. Most people wanted to legitimise symptoms and care needs before seeking help, but their relationship status and sexual behaviour also influenced their impetus to treat symptoms.

### Seeking control

These accounts provide insights into why symptoms reported in a research context might not be presented in a healthcare setting, especially an SHC. From their survey responses, 15 participants from our qualitative sample reported preferring the GP for hypothetical STI care, 7 would prefer SHCs, 4 opted for a contraception clinic and 1 person chose an internet site offering treatment as their preferred option (table 3). Perceiving a non-STI cause of symptoms directed participants away from SHCs exemplifying contextualised and rational help-seeking behaviour.

Individuals described shifting between the four emergent help-seeking strategies for symptoms, for example, escalating their response from normalising symptoms to attempting self-treatment before actively deciding to seek care and attending a specific service depending on the suspected cause and level of concern about the symptoms

experienced.[10] How painful and how quickly symptoms developed also influenced help-seeking responses. Overall, responses focused on seeking control over symptom experiences, enacted in different ways and with differing thresholds for accommodating symptoms and living with uncertainty. As information was readily available from a variety of sources, emotional reassurance was prioritised by most symptomatic individuals unless symptoms were severe.

### Data integration

Findings from the semistructured interviews help explain survey data about attendance patterns at SHCs and service preferences for STI care and genitourinary symptoms. By using different data from the same participants, we extend understanding of help-seeking behaviour for symptoms, enable more detailed interpretation of these data and strengthen conclusions about use of SHCs and offering service choice (table 4).

## DISCUSSION

We explored the high levels of non-attendance at SHCs reported in national survey data through follow-up semistructured interviews to understand help-seeking strategies for genitourinary symptoms. Our findings suggest that generally people did not seek care at SHCs in response to experiencing symptoms. GPs were the preferred provider in both survey and semistructured interviews, although younger people and those reporting symptoms were more likely to have attended a clinic recently. Lack of awareness or lack of choice of services available may have affected participants' preferences. Help-seeking focused on gaining control over symptoms through four responses: not seeking care; seeking information; seeking non-specialist care; and attending an SHC. Participants often segued between different help-seeking pathways. The nature of symptoms and previous care-seeking influenced help-seeking. Surprisingly, we did not find quantifiable gender differences in non-attendance at SHCs despite other work reporting women being more likely to attend healthcare.[23]

A sequential mixed methods design enabled us to elicit additional detail about attendance and use findings from each dataset to inform interpretations of the other. For example, Natsal-3 did not collect data about use of nonspecialist services, but interview data provided insight into decision making and different care-seeking pathways. Sampling interview participants from the Natsal-3 general population sample generated a non-patient sample, which enabled us to consider help-seeking independent of medical settings.[13] The sample size and sampling strategy of Natsal-3 resulted in the survey sample being broadly representative of the British population; therefore, we can assume estimates of non-attendance at clinics and service preferences are generalisable at the national level.

Genitourinary symptoms are non-specific and may not be indicative of STIs, which presents interesting

challenges to understand related help-seeking behaviour. We included a wide range of symptoms to capture different help-seeking responses. The time frames of the survey questions relating to symptoms and to SHC attendance were not the same—symptoms were asked about in the past month and SHC attendance in the past year. We therefore knew which participants had not sought care at a clinic when they were interviewed for Natsal-3 but had no quantitative data about their care-seeking intentions or outcomes. The cross-sectional design of Natsal-3 means that it is not possible to determine the causality of care-seeking behaviour. Semistructured interviews provided data on care-seeking decisions and outcomes. Natsal data collection takes place once a decade and offered an opportunity to nest a qualitative subsample within the main study. Due to the time taken to conduct initial survey analysis to inform qualitative data collection and funding constraints, there was a delay between survey and semistructured interview data collection. This delay resulted in high levels of participant attrition due to non-contactability; participants who took part may not reflect help-seeking behaviours observed in the survey; however, the time frame enabled longer term reflections on help-seeking and enabled us to identify the shift between different strategies over time. This enriched our analysis and helped us understand help-seeking priorities in the context of changes in participants' lives over time. Recall of specific symptomatic episodes varied depending on the nature of the symptoms and how significant they were to participants.[10] We framed this study in terms of sexual health, which may have primed participants to discuss their experience in the context of sex and STIs and silenced other explanations.

As our study was not dependent on service attendance to recruit participants, we took a broader perspective on help-seeking behaviour compared with studies that sample from a healthcare setting (eg, refs 5 8 24). We looked at individuals' behaviour and responses to experiencing symptoms, instead of relying exclusively on hypothetical constructs about intended behaviour. Many studies have found discrepancies between intention and behaviour. Our approach addressed some of these methodological issues. Our findings support those from similar studies using patient samples suggesting that previous attendance at an SHC makes subsequent visits more normal and acceptable,[5 8] but stigma remains a significant barrier to initial attendance.[6 25 26] Other Natsal-3 analyses found >70% of men and >85% of women with a prevalent STI perceived themselves as not at all at risk or not at very much risk of STIs.[27] Although increased STI risk perception was associated with increased STI healthcare use, mediation analysis suggested that risk perception was neither necessary nor sufficient for seeking care,[27] warranting a broader understanding of help-seeking to SHCs.

We used a sexual health framing for this study and focused on non-attendance at specialist SHCs. Other studies, such as Low et al[28] approached their research on gynaecological cancer symptoms from a general perspective by not disclosing their specific disease focus to participants. Like this study, they found examples of self-management and seeking legitimation of symptoms. From a public health perspective, non-attendance at SHCs following experience of genitourinary symptoms is a problem if, as a consequence, diagnosis and treatment are delayed. Considering help-seeking in the context of people's lives helps understand their priorities for health and healthcare and reasons for non-attendance.[13] Our findings about individuals' rationales for non-attendance are similar to those found in a study by Buetow[14] and include the narrowing gap between patient and professional knowledge (due to alternate information sources) and reluctance to share misfortune with others (leading to concealment and not seeking care).

We found four main help-seeking responses for genitourinary symptoms that help explain non-attendance, which have different implications for practice. First, not seeking care has implications for potential unmet need for STIs, other diseases and health issues. Maintaining broad provision of integrated sexual health services[29] ensures availability of healthcare without requiring specific care-seeking to specialist clinics. Developing interventions to normalise attendance and targeting specific issues around tendencies to normalise, conceal or dismiss symptoms may shift some individuals to pathways in to care. We suggest non-attendance be considered as part of the range of care-seeking responses and understood as rational according to individual's own reasoning, beliefs and priorities,[30] which are often overlooked by the public health community. Interventions that align with individuals' priorities are more likely to achieve public health outcomes, for example, using Accelerated Partner Therapy to remotely test and treat sex partners of patients diagnosed with chlamydia.[31] Encouraging 'bodily self-determination'[12] (p. 595) whereby healthcare professionals respect the healthcare decisions of patients who are competent to do so even if they disagree so as not to deter other forms of help-seeking is important to maintain relationships between individuals and healthcare services.

Second, seeking information showed participants' willingness to improve their understanding of their symptoms. Although experiential knowledge was often prioritised, making accurate information easily accessible and signposting to healthcare services could help expedite attendance. Additionally, development of an online clinical care pathway has been shown to meet the needs for the fully automated management of chlamydia[32] and appeals to young people[33] and may bridge the gap between searching for information using the internet and accessing healthcare. Third, genitourinary symptoms are often presented to other services (such as primary care and contraception clinics who can provide some testing and treatment options or signpost to SHCs). This suggests individuals are exercising their right to choose care that best suits their needs. There is good uptake and

acceptability of non-SHC care for genitourinary symptoms supporting policies to widen sexual health provision outside of specialist services[34]; this offers additional opportunities to test, treat and manage genitourinary symptoms, providing that healthcare professionals maintain sexual health skills. Effective signposting, communication and referrals between services will help timely management in the most appropriate service. Finally, delayed seeking to SHCs is associated with onward transmission of infection.[35] Although GPs are preferred initially, and participants were reluctant to go to an SHC, those who had attended specialist care had good experiences and would choose to reattend if needed. There is a disjuncture between anticipated and actual experiences of SHCs. Reducing barriers to access, including normalising attendance, is essential to ensure care-seekers do not experience further delays if they decide to seek specialist care.

Future surveys should examine intentions to seek care and a wider range of actual care-seeking outcomes for genitourinary symptoms to build on the exploratory findings of the qualitative strand of this study. Composite measures of unmet need combining risk behaviours, symptom experiences and STI testing and service use are needed to identify those with most need for healthcare and improve intervention targeting and service provision.

## CONCLUSION

Appropriate help-seeking in response to genitourinary symptoms helps ensure underlying care needs are met, reducing the burden of untreated infection and improving sexual and reproductive health. We found that the majority of participants who reported symptoms in Natsal-3 had not sought specialist help at an SHC; through qualitative interviews, we observed four main help-seeking strategies that explained the survey results. Overall, we can conclude that help-seeking occurs to regain control over physical symptoms and individuals prioritise emotional reassurance from a source that is accessible and familiar. The findings from this study are largely reassuring in that they suggest existing service provision across different types of healthcare settings in Britain provide sufficient choice and accessibility to high-quality care regardless of perceived cause. Integrated services, screening programmes and the expansion of self-testing provide opportunities to address untreated STI and unmet sexual health needs even if face-to-face care is not actively sought. However, recent progress is threatened by severe public health funding cuts that are already damaging the delivery of sexual healthcare in Britain.

**Acknowledgements** The authors would like to thank all of the participants of Natsal-3 and those who took part in the follow-up interviews for this study. Thank you also to Dr. Shema Tariq for conceptual help with integration of quantitative and qualitative data at an early stage of writing this paper. The authors benefitted from qualitative analysis input from Dr. Adam Bourne and members of the Kritikos Study Group at LSHTM.

**Contributors** FM conceived the idea for this study in collaboration with KW, CHM and the National Survey of Sexual Attitudes and Lifestyles (Natsal) team. FM designed the topic guide, conducted and analysed all of the qualitative interviews. FH and KW read and coded a subset of interviews. FM carried out all of the statistical analysis with the support of CHM and MJP. FM wrote the first draft of the paper and did all revisions based on feedback from all named authors (KW, CHM, KM, CT, SC, JD, NF, MJP and FH). FM was responsible for submitting the paper and all related correspondence, revising the manuscript and responding to reviewers comments and re-submission.

**Funding** This study was funded by the Economic and Social Research Council ES/J500021/1. Natsal-3 is a collaboration between University College London (London, UK), the London School of Hygiene & Tropical Medicine (London, UK), NatCen Social Research, Public Health England (formerly the Health Protection Agency) and the University of Manchester (Manchester, UK). The study was supported by grants from the Medical Research Council (G0701757) and the Wellcome Trust (084840), with contributions from the Economic and Social Research Council and the Department of Health. Since September 2015, KM has been core funded by the UK Medical Research Council and Chief Scientist Office (MC_UU_12017-11, SPHSU11).

**Competing interests** None declared.

**Patient consent for publication** Not required.

**Ethics approval** Natsal-3 was granted ethical approval by the National Research Ethics Service(NRES) Committee South Central – Oxford A (reference: 09/H0604/27). The qualitative strand of this study was given ethics approval by NRES Committee South Central – Oxford A 11/H0604/10 and London School of Hygiene & Tropical Medicine Observational/Interventions REC 6538. All participants included in this study gave written consent to participate.

**Provenance and peer review** Not commissioned; externally peer reviewed.

**Data availability statement** An anonymised dataset is available to academic researchers from the UK Data Service, https://discover.ukdataservice.ac.uk/; SN: 7799; persistent identifier: 10.5255/UKDA-SN-77991-1.

**ORCID iDs**
Fiona Mapp http://orcid.org/0000-0003-0733-6036
Clare Tanton http://orcid.org/0000-0002-4612-1858

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
