## [Reviewer comments · BMJ Open]

ARTICLE DETAILS

TITLE (PROVISIONAL)	Help-seeking for genito-urinary symptoms: A mixed methods study from Britain's Third National Survey of Sexual Attitudes and Lifestyles (Natsal-3)
AUTHORS	Mapp, Fiona; Wellings, Kaye; Mercer, Catherine H; Mitchell, Kirstin; Tanton, Clare; Clifton, Soazig; Datta, Jessica; Field, Nigel; Palmer, Melissa J; Hickson, Ford Colin Ian

VERSION 1 – REVIEW

REVIEWER	Emmanuel Morhe University of Health and Allied Sciences
REVIEW RETURNED	27-Apr-2019

GENERAL COMMENTS	Good job done
---------------

REVIEWER	Candy Wilson Uniformed Services University of the Health Sciences, Bethesda MD
REVIEW RETURNED	30-Apr-2019

GENERAL COMMENTS	Thank you for the submission of your paper titled, "Experiencing genito-urinary symptoms and not attending a sexual health clinic: Mixed Methods evidence from Britain's third National Survey of Sexual Attitudes and Lifestyles". Engaging patients in the appropriate health care system service is a key component when genitourinary symptoms arise. This paper attempts to describe many of the issues that contribute to health seeking behaviors and provides possible strategies to inform health care providers and policymakers that strive to make the sexual health clinic more appealing as a health care option. I found the paper confusing at times given the two competing outcomes (help-seeking and non-attendance in the SHC) were not clearly delineated in the results section. Further, several grammatical errors distracted me from the point the authors tried to convey. I will mention some of these sentences but recommend the authors seek guidance from an editor to assist with writing. There were several double and triple negatively stated sentences that were confusing. Define "Help-seeking" in the introduction. Please briefly describe the specialist make-up in the sexual health clinic as compared to the GP. In the introduction, explain for the non-British readers why help-seeking in the SHC is more important than help-seeking in general for these symptoms. Your study results informed me that patients desire a rapport with their provider when seeking care for very private health issues such as genitourinary symptoms.
---

	Title: Awkward. Consider rewriting. A possible option: Clinic choice for patients with genitourinary symptoms: A mixed methods study from Britain's Third National Survey of Sexual Attitudes and Lifestyles Keywords: in the manuscript text, (p. 2) the words listed seem appropriate. I am not sure what "GUM" is though. Abstract: first line of the conclusion has three negative words ("non-", "not", and "nothing"). Please reword to succinctly describe the meaning. Article Summary: The last limitation is a problem. Make sure you describe the reason for this delay for the interviews and how that could have impacted the participants responses. Page 3, line 8. Change one-fifth to 20% to remain parallel with your other statistic of 6% in the same sentence. Page 3, line 23. Consider: If left undiagnosed and untreated, the underlying disease can cause... Page 3, line 24. Consider: This lack of response to symptoms contributes... Page 4, line 5. Consider: Natsal-3 is a probability sample survey (n = 15,162) of sexual behavior among Britain women and men aged 16-74... Page 5 semi-structured interviews: I am concerned about the time delay between survey response and interviews (22 – 44 months). Please give a sentence or two for the delay and how this delay was managed in the analysis. Applying a "Interpretive Phenomenological Analysis", was bracketing used during the interviews and analysis? Data Integration. Great reference. Can you give more how it was applied in your study? Who was involved in the integration analysis? Table 1. By displaying this table first strikes me that your key objective is the "non-attendance" and the "help-seeking" is secondary. But, this is not a problem, just a comment. Page 8, line 25. "...five did not have English as their first language." How did the study team handle the language difference? Who interviewed? Were they fluent in the language? Was it part of the inclusion criteria? Table 2: OK Table 3: A nice way to synopsis the findings. Page 11, line 23. Reword the sentence, "Not seeking healthcare did not mean 'doing nothing' about symptoms; participants concealed symptoms, normalized them as physiological fluctuations or dismissal care needs." Consider: Specific behaviors and actions displayed by participants who chose not to seek care included normalizing, concealing, or dismissal of symptoms. Page 11, line 26. Reword sentence: STI stigma underpinned... Page 19, line 37-Pge 20, line 34 very well written and succinctly described your findings with relevant literature. Page 20. Add a conclusion section. I hope this review was helpful and I look forward to your future work.
--	--

REVIEWER	C Ooi Clinic 16 Royal North Shore Hospital NSW, Australia
REVIEW RETURNED	05-Jul-2019

GENERAL COMMENTS	mapp et al present and synthesise data of reports of GU symptoms and care seeking behaviour drawn from Natsal-3. From
---

	this cohort a subgroup were invited to participate in semistructured interviews. This is an interesting paper which reports health seeking behaviour and non-attendance at SHCs of people experiencing GU symptoms comparing age and sex. There are some items that need to be addressed. this paper questions why patients are not attending SHC, however after reading considers- where people who experience GU conditions (do) access care. semi structured interviews with non - SHC attenders do not seem to directly address why they did not attend a SHC; rather why they attended elsewhere.  1. some sentences require rewording for clarification and simplification. p2, ln 15-18; p2 ln 21-22,; 2. several inferences drawn in the conclusions and presented in the abstract are not borne out by the results. pl expand and clarify. p 2. ln 13-15; 24-26 preference for care may be dictated by lack of choice of service available. 3. strengths and limitations- recall bias is mentioned . expand on how semistructured interviewees were recruited; whether incentives were offered? geographical location? health care seeking behaviour may vary between rural and urban areas. 4. Further information is required regarding the services provided in England for an overseas audience. Are any fee based/ free/appointment or walk in? / geographical locations and availability/ hours of opening/ requirement to be a citizen/waiting times? there are many reasons that pts may chose one service over another? 5. only specific GU symptoms were sought- this should be made clear in the abstract. Further, urinary frequency was excluded as indicative of UTIs- this is also the case in female dysuria. consider excluding this measure also. Abdon=minal pain/pelvic pain may also be open to interpretation bias- and may not represent an accurate report of pain associated with STIs. for a stronger paper, only symptoms classical for STIs should be included eg male dysuria, urethral discharge, post coital bleeding, dyspareunia . 6. Do SHC have a particular triage policy that would exclude certain populations or particularly target certain populations- this will affect whether individuals will consider accessing care at these location 7. p11, ln 23-26. this sentence does not seem to make sense. surely concealing symptoms, dismissing care needs and normalisation is essentially 'doing nothing' about symptoms? 8. no perception of STI risk presented from interviews- this data would be important to establish participants' perceived needs please add. further to this, authors report that familiarity with symptoms reduced health seeking- this may represent long standing conditions for which pts have sought care previously- this should be acknowledged in the text. 9. p12, ln 27- normalising symptoms- this paragraph would be strengthened with examples.
--	--

	10. in discussion- please clarify the expected roles of the services discussed- SHC, GP, Family planning etc to assist an international audience in understanding the intersection and divisions of care provision.
--	---

VERSION 1 – AUTHOR RESPONSE

Reviewer reports:

Reviewer 1: Emmanuel Morhe

Comment 1: Good job done

Response: Thank you

Reviewer 2: Candy Wilson

Thank you for the submission of your paper titled, “Experiencing genito-urinary symptoms and not attending a sexual health clinic: Mixed Methods evidence from Britain’s third National Survey of Sexual Attitudes and Lifestyles”. Engaging patients in the appropriate health care system service is a key component when genitourinary symptoms arise. This paper attempts to describe many of the issues that contribute to health seeking behaviors and provides possible strategies to inform health care providers and policymakers that strive to make the sexual health clinic more appealing as a health care option. I found the paper confusing at times given the two competing outcomes (help-seeking and non-attendance in the SHC) were not clearly delineated in the results section. Further, several grammatical errors distracted me from the point the authors tried to convey. I will mention some of these sentences but recommend the authors seek guidance from an editor to assist with writing. There were several double and triple negatively stated sentences that were confusing.

Define “Help-seeking” in the introduction. Please briefly describe the specialist make-up in the sexual health clinic as compared to the GP. In the introduction, explain for the non-British readers why help-seeking in the SHC is more important than help-seeking in general for these symptoms. Your study results informed me that patients desire a rapport with their provider when seeking care for very private health issues such as genitourinary symptoms.

Response: We have added in a definition of help-seeking and linked it to more explanation of services in the UK (page 2, lines 27-30).

Title: Awkward. Consider rewriting. A possible option: Clinic choice for patients with genitourinary symptoms: A mixed methods study from Britain’s Third National Survey of Sexual Attitudes and Lifestyles

Response: Changed to “Help-seeking for genito-urinary symptoms: A mixed methods study from Britain’s Third National Survey of Sexual Attitudes and Lifestyles (Natsal-3)” to encompass the broader aim of understanding help-seeking not just clinic attendance

Keywords: in the manuscript text, (p. 2) the words listed seem appropriate. I am not sure what “GUM” is though.

Response: I have spelt out the acronym – Genito-Urinary Medicine Clinics.

Abstract: first line of the conclusion has three negative words (“non-“, “not“, and “nothing”). Please reword to succinctly describe the meaning.

Response: We have re-written a concise conclusion.

Article Summary: The last limitation is a problem. Make sure you describe the reason for this delay for the interviews and how that could have impacted the participants responses.

Response: We have added in the context for the delay in the article summary and expanded the description of this limitation in the discussion (page 19, lines 31-33 and page 20 lines 1-7).

Page 3, line 8. Change one-fifth to 20% to remain parallel with your other statistic of 6% in the same sentence.

Response: We have changed this as recommended.

Page 3, line 23. Consider: If left undiagnosed and untreated, the underlying disease can cause...

Response: We have changed this as suggested.

Page 3, line 24. Consider: This lack of response to symptoms contributes...

Response: We have changed this as suggested.

Page 4, line 5. Consider: Natsal-3 is a probability sample survey (n = 15,162) of sexual behavior among Britain women and men aged 16-74...

Response: We have left this sentence as it is as the survey was not only of British men and women....it also included those who were resident here at the time of data collection who may not have been British.

Page 5 semi-structured interviews: I am concerned about the time delay between survey response and interviews (22 – 44 months). Please give a sentence or two for the delay and how this delay was managed in the analysis. Applying a “Interpretive Phenomenological Analysis”, was bracketing used during the interviews and analysis?

Response: We have added in more information about the delay (page 5, lines 16-18) which is complemented by further discussion about the impact of the delay in the discussion (page 19, lines 31-33 and page 20 lines 1-7).. We did not use bracketing as this is more closely linked to Husserl’s branch of phenomenology. We employed techniques to encourage reflexivity throughout the study such as writing field notes and double coding and discussion of themes. We have made these practices more explicit in the methods.

Data Integration. Great reference. Can you give more how it was applied in your study? Who was involved in the integration analysis?

Response: Thank you. We have added more detail about the integration process.

Table 1. By displaying this table first strikes me that your key objective is the “non-attendance” and the “help-seeking” is secondary. But, this is not a problem, just a comment.

Response: We have added in an additional sentence at the beginning of the results to explain how data are presented.

Page 8, line 25. “...five did not have English as their first language.” How did the study team handle the language difference? Who interviewed? Were they fluent in the language? Was it part of the inclusion criteria?

Response: We have added in clarification of this aspect, page 9 line 13-14.

Table 2: OK

Table 3: A nice way to synopsis the findings.

Response: Thank you

Page 11, line 23. Reword the sentence, “Not seeking healthcare did not mean ‘doing nothing’ about symptoms; participants concealed symptoms, normalized them as physiological fluctuations or

dismissal care needs.” Consider: Specific behaviors and actions displayed by participants who chose not to seek care included normalizing, concealing, or dismissal of symptoms.

Response: We have amended this sentence to simplify the message.

Page 11, line 26. Reword sentence: STI stigma underpinned...

Response: We have reworded this.

Page 19, line 37-Pge 20, line 34 very well written and succinctly described your findings with relevant literature.

Response: Thank you

Page 20. Add a conclusion section.

Response: We have added a conclusion and think it helps to bring the article together. Thank you for this suggestion.

I hope this review was helpful and I look forward to your future work.

Response: Thank you for taking the time to provide this helpful review.

Reviewer 3: C Ooi

Comment 1: mapp et al present and synthesise data of reports of GU symptoms and care seeking behaviour drawn from Natsal-3. From this cohort a subgroup were invited to participate in semistructured interviews. This is an interesting paper which reports health seeking behaviour and non-attendance at SHCs of people experiencing GU symptoms comparing age and sex. There are some items that need to be addressed.

this paper questions why patients are not attending SHC, however after reading considers- where people who experience GU conditions (do) access care. semi structured interviews with non -SHC attenders do not seem to directly address why they did not attend a SHC; rather why they attended elsewhere.

Response: We have updated the title of the paper as recommended by reviewer 2 to better reflect the focus on help-seeking strategies not just non-attendance. We have added an additional sentence at the beginning of the results to make it clearer how we used the survey and interview data to quantify non-attendance at SHCs and then look to explain the reasons and investigate help-seeking more generally.

1. some sentences require rewording for clarification and simplification. p2, ln 15-18; p2 ln 21-22,;

Response: We have amended the abstract to make it easier to read and understand.

2. several inferences drawn in the conclusions and presented in the abstract are not borne out by the results. pl expand and clarify. p 2. ln 13-15; 24-26. preference for care may be dictated by lack of choice of service available.

Response: As above, we have amended the abstract to better link results and conclusions and noted influences on preference in the discussion.

3. strengths and limitations- recall bias is mentioned . expand on how semistructured interviewees were recruited; whether incentives were offered? geographical location? health care seeking behaviour may vary between rural and urban areas.

Response: We have re-written this section in line with comments from reviewer 2 to improve clarity and accuracy. We have also added more detail to the methods section describing the sampling of interviewees.

4. Further information is required regarding the services provided in England for an overseas audience. Are any fee based/ free/appointment or walk in? / geographical locations and availability/ hours of opening/ requirement to be a citizen/waiting times? there are many reasons that pts may chose one service over another?

Response: We have added more detail as suggested about SHC fees, appointments, geographical locations and accessibility (page 3 lines 1-9) however due to the devolved nature of the commissioning of SHCs, and lack of recent data it is not possible to provide more information on opening hours and waiting times.

5. only specific GU symptoms were sought- this should be made clear in the abstract. Further, urinary frequency was excluded as indicative of UTIs- this is also the case in female dysuria. consider excluding this measure also. Abdominal pain/pelvic pain may also be open to interpretation bias- and may not represent an accurate report of pain associated with STIs. for a stronger paper, only symptoms classical for STIs should be included eg male dysuria, urethral discharge, post coital bleeding, dyspareunia .

Response: We have clarified that we looked at specific GU symptoms in the abstract. We took clinical advice from the wider Natsal-3 team (as now stated in Box 1, page 5) who developed and piloted the questions to exclude passing urine more often than usual but retain the other symptoms. These symptoms are routinely asked about in sexual health clinic triage forms and as part of a sexual health consultation (also added to the paper pg 4 line 19-20) therefore we feel are relevant to this paper about help-seeking at SHCs. The non-specificity of symptoms is an interesting aspect of studying help-seeking for STIs and we wanted to capture help-seeking across a wide range of symptom experiences (added to the discussion pg. 19 lines 23-25). We acknowledge the non-specificity of GU symptoms but this in itself is a factor of help-seeking and part of the decision making about which service to use.

6. Do SHC have a particular triage policy that would exclude certain populations or particularly target certain populations- this will affect whether individuals will consider accessing care at these location

Response: Funding cuts have led to asymptomatic patients being managed through online pathways (mentioned in the introduction, page 3 line 8) ie requesting STI self-sampling kits instead of being seen in clinic. Some clinics offer dedicated clinics for young people, MSM, sex workers etc but this is down to local commissioned services and there is no data available on how common this is.

7. p11, ln 23-26. this sentence does not seem to make sense. surely concealing symptoms, dismissing care needs and normalisation is essentially 'doing nothing' about symptoms?

Response: We have reworded this sentence. Clinically it may look like patients are 'doing nothing' but sociologically there is a lot of work going on to manage symptoms and this is what we present.

8. no perception of STI risk presented from interviews- this data would be important to establish participants' perceived needs please add. further to this, authors report that familiarity with symptoms reduced health seeking- this may represent long standing conditions for which pts have sought care previously- this should be acknowledged in the text.

Response: We did not ask about STI risk in qualitative interviews as this was not the focus of the study and the literature suggests poor links between perceived and actual STI risk. We have added information cited in another Natsal-3 paper about STI risk and healthcare use to the discussion. Thank you for highlighting this absence. We have acknowledged the link between familiarity of long term conditions and reduced healthcare-seeking as suggested.

9. p12, ln 27- normalising symptoms- this paragraph would be strengthened with examples.

Response: We have added in quotes to support the points made.

10. in discussion- please clarify the expected roles of the services discussed- SHC, GP, Family planning etc to assist an international audience in understanding the intersection and divisions of care provision.

Response: We feel with the additional information about services in the introduction and a few tweaks to the discussion is sufficient to understand our findings in the context of the British healthcare system.

Thank you for your helpful review.

VERSION 2 – REVIEW

REVIEWER	Candy Wilson Uniformed Services University of the Health Sciences United States
REVIEW RETURNED	21-Sep-2019
GENERAL COMMENTS	nice response